# Risk Factors for Anticancer Drug-Induced Hyponatremia: An Analysis Using the Japanese Adverse Drug Report (JADER) Database

**DOI:** 10.3390/medicina59010166

**Published:** 2023-01-13

**Authors:** Naohisa Tamura, Tomoaki Ishida, Kei Kawada, Kohei Jobu, Shumpei Morisawa, Saburo Yoshioka, Mitsuhiko Miyamura

**Affiliations:** 1Graduate School of Integrated Arts and Sciences, Kochi University, Kohasu, Oko-Town, Nankoku City 783-8505, Kochi, Japan; 2Department of Pharmacy, Kochi Medical School Hospital, 185-1 Kohasu, Oko-Town, Nankoku City 783-8505, Kochi, Japan

**Keywords:** hyponatremia, cisplatin, lung cancer, Japanese Adverse Drug Event Report Database

## Abstract

*Background and Objectives:* Hyponatremia is among the most prevalent electrolyte abnormalities observed in patients with cancer during chemotherapy. Therefore, managing hyponatremia is crucial since it causes a severe electrolyte imbalance that can lead to significant mortality, and this study aimed to investigate the relationship between hyponatremia, anticancer drugs, and cancer types. *Materials and Methods:* Reported odds ratios were calculated and evaluated based on adverse event reports submitted to the Japanese Adverse Drug Event Report (JADER) database. *Results:* Overall, 2943 patients had hyponatremia. Notably, cisplatin, pemetrexed, and etoposide had marked hyponatremia signals. In addition, significant hyponatremia signals were detected for oesophageal, lung, and renal cancers. *Conclusions:* Hyponatremia has been reported in women and patients with lung cancer receiving cisplatin, with a growing trend in the number of elderly patients receiving cisplatin. Furthermore, since the onset of hyponatremia during cisplatin administration is frequently reported within 10 days, patient information should be thoroughly examined before and monitored throughout the administration, which can contribute to the early detection and prevention of hyponatremia.

## 1. Introduction

Hyponatremia is an important electrolyte imbalance associated with high morbidity and mortality [1]. Normal serum sodium concentrations generally range between 136 and 143 mEq/L, with hyponatremia defined as less than 135 mEq/L. As hyponatremia worsens, the risk of physical instability, falls, and inattention increases, including central nervous system symptoms, such as impaired consciousness and seizures caused by cerebral edema and increased intracranial pressure, may occur [2]. These effects reduce the patient’s quality of life and interfere with treatment. In more severe cases, symptoms such as headache, lethargy, and jaundice may occur, eventually leading to seizures, coma, and death [3]. The incidence of hyponatremia in hospitalized patients is reported to be 15–30% [4,5] and is known to be higher in the elderly with cirrhosis and heart failure [6,7].

Hyponatremia occurs when the total body water content is excessive in relation to the total sodium content in the body. Common causes of hyponatremia include medications and antidiuretic hormone (ADH) deficiency syndrome. Also, some drugs can induce hyponatremia, especially anticancer agents [8]. Hyponatremia due to ADH incompatibility syndrome or anticancer agents is often associated with normal extracellular fluid volume. Accordingly, the status of extracellular fluid volume (decreased, normal, or increased) should be considered in hyponatremia because it reflects the total sodium in the body. Although hyponatremia associated with the use of certain anticancer agents has been reported in previous studies, re-reported cases are limited because hyponatremia due to anticancer agents is rare [9,10,11]; therefore, vast knowledge of hyponatremia due to anticancer agents is limited. Further, symptoms of hyponatremia are similar to those of chemotherapy side effects, such as fatigue, nausea, and somnolence, among which the symptoms of hyponatremia may be missed. Therefore, it is important to anticipate the occurrence of hyponatremia and to treat it appropriately in order to avoid adverse effects.

The analysis was conducted on several cases using the Japanese Adverse Drug Event Report (JADER) database. For post-marketing pharmacovigilance of drugs, such databases enable many cases of adverse events (AEs) to be collected and analyzed using pharmacoepidemiological methods to assess the presence and strength of a causal relationship. Many countries have established voluntary AE reporting systems to detect unknown AEs, which are used to collect case reports of suspected adverse drug reactions at the national level. The JADER database obtains information on cases of drug reactions, which are reported voluntarily by the Pharmaceuticals and Medical Devices Agency (PMDA). Rare AEs are often detected only after a drug has been marketed and used in a large number of patients. Spontaneous reporting databases can help identify rare AEs. Some previous studies have used JADER to investigate rare adverse effects [12,13,14]. Therefore, using JADER, it is possible to investigate the relationship between hyponatremia and each anticancer agent, as well as potential risk factors related to patient characteristics.

This study aimed to evaluate the relationship between anticancer drugs and hyponatremia using the reporting odds ratio (ROR) and examine the association between hyponatremia and anticancer agents and patient characteristics. We also analyzed the time to onset of hyponatremia associated with anticancer drugs.

## 2. Materials and Methods

### 2.1. Study Design

The data recorded from April 2004 to November 2020 in the JADER database were downloaded from the PMDA website (http://www.pmda.go.jp/) (accessed on 1 April 2021). The JADER dataset comprises four tables containing the following information: (1) patient information, which includes sex, age, and body weight; (2) patient’s drug history; (3) patient adverse events and outcomes; and (4) medical history and primary illness. Subsequently, the four tables were integrated using the FUND E-Z Backup Archive (FUND E-Z Development Corporation, NY, USA). Patients whose age and sex were unclear were excluded. Further, in the patient’s drug history, the contribution of drug-related AEs was categorized into three codes as follows: “suspected drug,” “concomitant drug,” and “interaction.” The category of “suspected drugs” was examined in this study. Overall, 14 anticancer drugs approved in Japan (cisplatin, carboplatin, oxaliplatin, cyclophosphamide, pemetrexed, vincristine, etoposide, nivolumab, pembrolizumab, irinotecan, doxorubicin, paclitaxel, docetaxel, and gemcitabine) were selected as the drugs of interest in this study; these anticancer drugs were in more than 10 reports of hyponatremia. First, crude RORs were calculated to evaluate the effect of anticancer drugs on the onset of drug-induced hyponatremia. Second, crude RORs were also calculated to assess the impact of cancer type at the onset of drug-induced hyponatremia. Third, the adjusted RORs using a multivariate analysis were calculated for the reporting year, age, sex, cancer type, and anticancer drugs to analyze potential hyponatremia-associated factors. Finally, we examined the onset time of cisplatin-associated hyponatremia using the Weibull distribution.

### 2.2. Definitions of Side Effects and Medication

The International Conference on Harmonization of Technical Requirements for Registration of Pharmaceuticals for Human Use Medical Dictionary for Regulatory Activities ver. 24.1 was used to extract the AEs and underlying diseases recorded in the JADER database. Details of the definitions of AEs and underlying diseases are provided in the Appendix A.

### 2.3. Analysis of the ROR

The calculation method for the crude ROR is outlined in this section. First, the cases were categorized into groups (a) to (d), as follows:

(a) Individuals who received the drugs and displayed AEs; (b) individuals who received the drugs but did not show any AEs; (c) individuals who did not receive the drugs and had AEs; (d) individuals who did not receive the drugs and did not show any AEs. Next, the crude ROR was calculated using the following method:Crude ROR = (a/b)/(c/d) 
$$95\%\;\mathrm{confidence}\;\mathrm{interval}\;(\mathrm{CI})=\exp\;[\log(\mathrm{ROR})\;\pm \;1.96\;\sqrt{\left(1/a\right)+\left(1/b\right)+\left(1/c\right)+\left(1/d\right)}]$$

A signal was detected when the lower limit of 95% CI of the crude ROR exceeded 1. RORs were expressed as point estimates with 95% Cis, and data were analyzed using Fisher’s exact test.

We calculated the adjusted ROR following previous reports [15,16]. In addition, the cases were stratified by age as follows: 0–59 and ≥60 years. The adjusted ROR was calculated using the 0–59-year age group as the control group. Furthermore, sex (female), year of reporting, and stratified age groups were coded to construct the logistic model; the following logistic model was used for the analysis:Log (odds) = β_0_ + β_1_Y + β_2_S + β_3_A + β_4_D + β_5_C + β_7_S * D + β_8_A * D + β_9_C * D
Y = reporting year, S = sex, A = stratified age group, D = drug, and C = cancer type.

Results were considered statistically significant at *p* ≤ 0.05. Similarly, results were considered statistically significant for the analysis of interactions at *p* ≤ 0.10. These analyses were performed using JMP 14.0 (SAS Institute, Cary, NC, USA).

### 2.4. Analysis of Onset of Adverse Events

The onset of AEs was deduced from when the patient received their first prescription for hyponatremia. Records with complete hyponatremia occurrence and prescription start date were used for the time-to-onset analysis. Therefore, it is crucial to consider the right truncation when evaluating the time-to-onset of hyponatremia. We determined an analysis period of 365 days after the start of administration [17]. The median duration, quartiles (interquartile range [IQR]), and Weibull shape parameters (WSPs) were used to assess the onset data. Additionally, the scale parameter α of the Weibull distribution determines the scale of the distribution function. Specifically, a larger-scale value (α) stretches the distribution, whereas a smaller-scale value (α) shrinks it. WSP β of the Weibull distribution determines the shape of the distribution function. Furthermore, larger and smaller shape values produced both left- and right-skewed curves. The WSP β of the Weibull distribution was employed to show the hazard level over time without a reference population. When β = 1, the hazard is estimated to be constant over time. In contrast, when β > 1 and the 95% CI of β excludes 1, the hazard increases with time [18]. These analyses were performed using JMP 14.0 (SAS Institute, Cary, NC, USA).

This research uses only existing unlinkable anonymized data and does not require ethical review by an ethics committee.

## 3. Results

### 3.1. Crude RORs and the Number of Cases of Each Anticancer Drug Associated with Hyponatremia

Among the 595,121 patients reported between April 2004 and December 2020, 2943 developed hyponatremias. Therefore, the crude ROR (95% CI) of each anticancer drug linked with hyponatremia in the JADER database was calculated. Notably, the following three drugs were detected in the ROR analysis of significant signals: cisplatin (ROR: 4.19, 95% CI: 3.64–4.82), pemetrexed (ROR: 2.85, 95% CI: 2.09–3.80), and etoposide (ROR: 1.73, 95% CI: 1.27–2.32) (Table 1).

### 3.2. Crude RORs and the Number of Cases of Each Cancer Site Associated with Hyponatremia

The crude ROR (95% CI) for each cancer site related to hyponatremia in the JADER database was calculated.

Significant signals were detected for esophageal (ROR: 2.20, 95% CI: 1.25–3.47), lung (ROR: 1.80, 95% CI: 1.70–2.40), and renal (ROR: 1.50, 95% CI: 1.16–1.99) cancers (Table 2).

### 3.3. Analysis of Adjusted ROR for the Patient Character with Hyponatremia

Table 3 shows the adjusted RORs and 95% CIs. In the multivariate logistic regression analysis, significant contributions were observed for females (ROR: 1.33, 95% CI: 1.24–1.44, *p* < 0.001), age (≥60 years, ROR: 2.73, 95% CI: 2.48–3.00, *p* < 0.001), cancer site (lung, ROR: 1.49, 95% CI: 1.25–1.78, *p* < 0.001; renal, ROR: 1.57, 95% CI: 1.23–2.01, *p* < 0.001), and the use of cisplatin (ROR: 4.05, 95% CI: 3.47–4.73, *p* < 0.001) (Table 3). Moreover, significant (*p* < 0.0001) interactions were observed between age (≥60 years) and sex (male). Similarly, the interactions between cisplatin and sex (female) (*p* < 0.001), cisplatin use, and age (≥60 years) (*p* = 0.053) were significant.

### 3.4. The Onset of Hyponatremia in Patients Treated with Cisplatin

Figure 1 shows the time-to-onset profiles of hyponatremia following cisplatin administration. Overall, 221 patients developed hyponatremia with cisplatin as the suspected drug. Of these, the start and onset dates of hyponatremia were clearly shown in 162 patients. In addition, the onset of hyponatremia was analyzed using the Weibull distribution. The scale parameters were α = 10.38, 95% CI: 8.77–12.26; β = 0.98, 95% CI: 0.89–1.08; and the median of onset day was (7, IQR: 5–8). Furthermore, the onset of hyponatremia occurred after the first day of cisplatin administration, and 86% of the patients developed this complication within 10 days of the start of administration.

It showed the time-to-onset profiles of hyponatremia following cisplatin administration. The onset of hyponatremia was analyzed using the Weibull distribution, wherein the scale parameter α of the Weibull distribution determines the scale of the distribution function. The WSP β of the Weibull distribution was employed to show the hazard level over time without a reference population.

## 4. Discussion

We discovered that the anticancer drugs (cisplatin, etoposide, and pemetrexed) and cancer types (esophageal, lung, and renal cancers) had hyponatremia signals in the JADER database. In addition, hyponatremia signals were detected in women and patients aged >60 years. Furthermore, the onset of hyponatremia, which is associated with cisplatin treatment, developed relatively early (within 10 days). Therefore, these findings may help reduce the risk of hyponatremia during chemotherapy to provide better anticancer therapy.

In this study, the anticancer drugs, which include cisplatin, etoposide, and pemetrexed, induced hyponatremia signals. The association between cisplatin and hyponatremia has been previously reported [19]. In contrast, reports of hyponatremia in patients administered pemetrexed and etoposide were not found. However, pemetrexed and etoposide were independent risk factors in the multivariate analysis (Table 3), implying that these anticancer drugs may also be related to hyponatremia. A study report showed that anticancer drugs produce vasopressin and may also affect diuresis [20]. Notably, some anticancer drugs induce vasopressin by altering the normal osmotic regulation of ADH secretion, which is induced by stimulating arginine vasopressin (AVP) secretion [20]. These anticancer drugs (cisplatin, etoposide, and pemetrexed) may also demonstrate these effects. Additionally, cisplatin stimulates AVP secretion; however, it may also directly damage the renal tubules’ interference with sodium reabsorption, which may result in hyponatremia. Also, oxaliplatin is a useful anticancer agent for esophageal cancer and is used worldwide [21]. However, in Japan, the indication for oxaliplatin for esophageal cancer was added in 2019, and the number of cases used is still small. Therefore, the analysis could not be performed due to the small number of reports in JADER, which is an adverse reaction report database in Japan. Therefore, it will be necessary to investigate the effects of these anticancer drugs in triggering hyponatremia in future studies [22].

Previous studies have reported the association between hyponatremia and lung, esophageal, head and neck, and urothelial cancers [20]. In this study, renal cancer had a hyponatremic signal, and patients with hyponatremia who had renal cancer have a higher mortality rate than those with normal serum sodium levels; therefore, its incidence should be prevented [23]. The mechanisms underlying hyponatremia developing in patients with renal cancer remain unknown. However, treatment with cisplatin is not typically used in patients with renal cancer [24]. Furthermore, molecularly targeted agents may be linked to hyponatremia since patients with renal cancer have been reported to develop hyponatremia after treatment with molecular therapies [25].

Our study detected a hyponatremia signal only with cisplatin in a multivariate analysis that considered sex, age, and cancer type. In addition, an interaction was detected between cisplatin and lung cancer, implying that consideration may be needed to monitor hyponatremia when using cisplatin in patients with lung cancer [17,25]. Further, in our study, Table 2 presents a higher significant OR difference in esophageal cancer; however, multivariate analysis depicted a marked decrease in the OR of esophageal cancer. Thirteen of the 17 patients with esophageal cancer were over 60 age, and 15 patients received cisplatin in the multivariate analysis. Therefore, age (over 60 age) and cisplatin administration might have contributed to the decreased OR of esophageal cancer in the multivariate analysis.

The anticancer drug cisplatin is classified as a highly emetogenic drug [26]. Previous reports have proposed that the risk factors for nausea are higher in females than in males [27,28]. High nausea levels can cause fluctuations in food intake and may result in electrolyte abnormalities. Additionally, the stimulation of the vomiting center was reported to promote ADH secretion, which may influence hyponatremia [29]. This study’s results propose that the female sex may influence hyponatremia; thus, hyponatremia associated with cisplatin may be affected by sex.

However, the number of reports of hyponatremia associated with cisplatin increased in patients aged >60 years, which may be due to reduced sodium reabsorption linked to age-related decline in renal function [30]. Additionally, moisture load during cisplatin administration may cause hyponatremia in the elderly [31].

Time-to-onset analysis provided new insights into the timing of hyponatremia onset. Cisplatin-induced hyponatremia is often observed within 10 days of administration. We proposed that cisplatin-induced impairment of electrolyte reabsorption from Henle’s hoof of the kidney was responsible for hyponatremia. Furthermore, increased urinary sodium excretion, high water reabsorption in the collecting ducts, and direct enhancement of ADH secretion due to impaired electrolyte reabsorption in Henle’s hoof are considered potential causes. Moreover, reports suggested that water loading to prevent renal injury during cisplatin administration may increase circulating blood volume and decrease serum osmolality [18]. Therefore, hyponatremia may occur early in cisplatin administration.

This study had some limitations. First, no evidence exists showing that the reported events are attributed to the drug. Second, gastrointestinal symptoms and food intake due to adverse drug reactions may vary according to the patient’s condition. Third, cases in the JADER database were reported spontaneously; therefore, reporting bias may exist. Fourth, JADER does not provide the clinical information necessary to diagnose adverse events and underlying diseases, such as laboratory values. Thus, in this study, we defined the diseases from the names of diseases reported spontaneously using the Dictionary of Medical Science ver. 24.1. Finally, the study’s data may represent a different patient population than that typically existing in clinical practice. Therefore, this study’s results should be further examined in cohort studies and randomized controlled trials.

## 5. Conclusions

This study suggested that the anticancer drugs (cisplatin, etoposide, and pemetrexed) and cancer types (esophageal, lung, and renal cancers) had hyponatremia signals, and hyponatremia may more likely occur in patients with lung cancer and those who were administered cisplatin. In addition, hyponatremia, which is associated with cisplatin treatment, was detected in women and patients aged >60 years. Furthermore, the onset of hyponatremia, which is associated with cisplatin treatment, developed relatively early (within 10 days). Based on this information, patient information should be carefully considered before administration, monitoring during treatment, and appropriate measures should be applied for patients requiring early sodium supplementation.

## Figures and Tables

**Figure 1 medicina-59-00166-f001:**
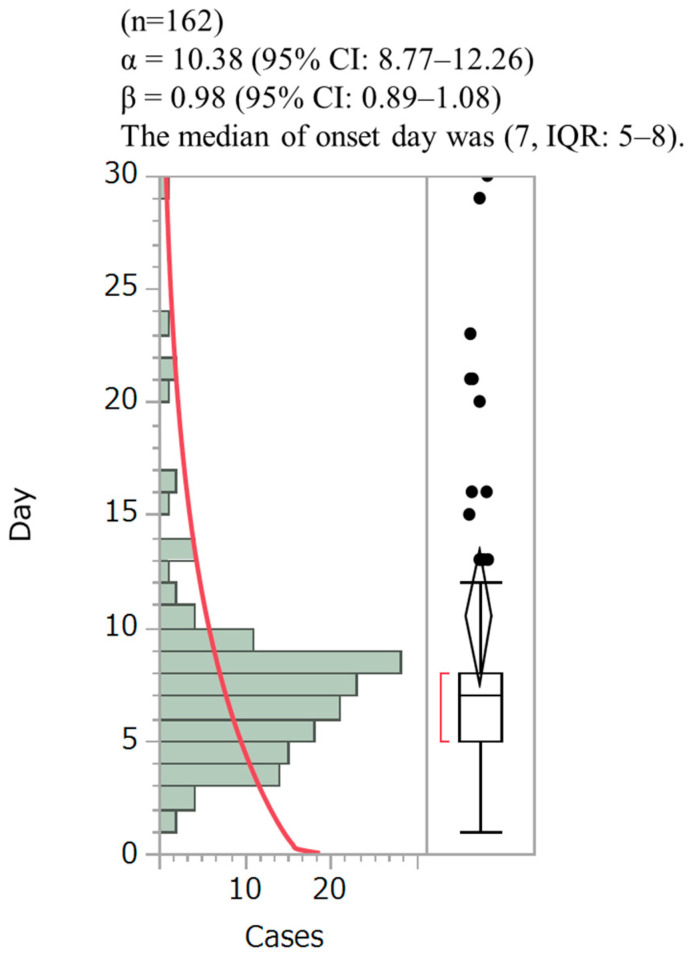
The number of hyponatremia cases associated with cisplatin by onset time in the Japanese Adverse Drug Event Report (JADER) database.

**Table 1 medicina-59-00166-t001:** The number of reported cases and crude ROR (95% CI) of hyponatremia based on anticancer drugs.

Drugs	Total (n)	Case (n)	Non-Case (n)	Ratio (%)	Crude ROR (95% CI)	*p*-Values
Total	595,121	2943	592,178	0.49		
Cisplatin	11,468	221	11,247	1.93	4.19 (3.64–4.82)	<0.001
Pemetrexed	3471	48	3423	1.38	2.85 (2.09–3.80)	<0.001
Etoposide	5307	45	5262	0.85	1.73 (1.27–2.32)	<0.001
Carboplatin	9266	56	9210	0.60	1.22 (0.92–1.60)	0.14
Vincristine	6179	34	6145	0.55	1.11 (0.77–1.56)	0.52
Pembrolizumab	5505	30	5475	0.54	1.10 (0.74–1.58)	0.56
Nivolumab	9307	49	9258	0.53	1.07 (0.79–1.41)	0.65
Cyclophosphamide	10,889	54	10,835	0.50	1.00 (0.75–1.31)	0.95
Docetaxel	7848	39	7809	0.50	1.00 (0.71–1.37)	0.94
Paclitaxel	11,161	54	11,107	0.48	0.98 (0.73–1.28)	0.95
Doxorubicin	6555	31	6524	0.47	0.96 (0.65–1.36)	0.92
Irinotecan	8695	41	8654	0.47	0.95 (0.68–1.30)	0.82
Gemcitabine	5338	21	5317	0.39	0.79 (0.49–1.22)	0.33
Oxaliplatin	11,973	27	11,946	0.23	0.45 (0.30–0.66)	<0.001

ROR, reporting odds ratio; JADER, Japan Adverse Drug Event Report; 95% CI, 95% confidence interval.

**Table 2 medicina-59-00166-t002:** The number of reported cases and crude ROR (95% CI) of hyponatremia according to the cancer site.

Cancer Site	Total (n)	Case (n)	Non-Case (n)	Ratio (%)	Crude ROR (95% CI)	*p*-Values
Total	595,121	2,943	592,178	0.49		
Esophageal	1607	17	1590	1.06	2.20 (1.25–3.47)	0.032
Lung	15,256	138	15,118	0.90	1.80 (1.70–2.40)	<0.001
Renal	7706	58	7648	0.75	1.50 (1.16–1.99)	0.002
Stomach	8202	52	8150	0.63	1.30 (0.96–1.70)	0.080
Prostate	7944	41	7903	0.52	1.00 (0.75–1.42)	0.75
Colorectal	15,913	50	15,863	0.31	0.63 (0.46–0.83)	<0.001
Breast	12,604	18	12,586	0.14	0.30 (0.17–0.45)	<0.001

ROR, reporting odds ratio; 95% CI, 95% confidence interval.

**Table 3 medicina-59-00166-t003:** The number of reported cases and ROR (95% CI).

	Total (n)	Case (n)	*p*-Values	Adjusted ROR (95% CI)	*p*-Values
Total	595,121	2943			
Reporting year	—	—	—	0.99 (0.99–1.00)	0.061
Sex, Female	289,118	1574	<0.001	1.33 (1.24–1.44)	<0.001
Age					
≥60 years	378,055	2430	<0.001	2.73 (2.48–3.00)	<0.001
Cancer site					
Renal	7706	58	0.002	1.57 (1.23–2.01)	<0.001
Lung	15,256	138	<0.001	1.49 (1.25–1.78)	<0.001
Esophageal	1607	17	0.003	1.06 (0.65–1.74)	0.17
Drugs					
Cisplatin	11,468	221	<0.001	4.05 (3.47–4.73)	<0.001
Etoposide	5307	45	<0.001	1.23 (0.85–1.58)	0.19
Pemetrexed	3471	48	<0.001	1.16 (0.85–1.58)	0.36
Interaction					
Cisplatin and Lung	—	—	—	1.50 (1.00–2.26)	0.051
Cisplatin and Renal	—	—	—	1.18 (0.16–8.84)	0.87
Cisplatin and Female	—	—	—	0.98 (0.74–1.31)	0.91
Cisplatin and ≥age 60	—	—	—	0.71 (0.50–1.00)	0.053

ROR, reporting odds ratio; 95% CI, 95% confidence interval.

## Data Availability

The datasets generated and analyzed during the current study are available from the corresponding author upon reasonable request.

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
