# Peer review of "Risk Factors for Anticancer Drug-Induced Hyponatremia: An Analysis Using the Japanese Adverse Drug Report (JADER) Database"

_medicina, 2023, doi:10.3390/medicina59010166_

Round 1

Reviewer 1 Report

This study address an important issue of chemotherapy induced hyponatremia. As acknowledged by the authors themselves, there are some limitations. 

Additionally, In my opinion, the readership would be interested to know more about the ROR of cisplatin and Oxaliplatin in table 1. It is well known that either Cisplatin or Oxaliplatin can be used (with capecitabine) for esophageal cancer chemotherapy. And Oxaliplatin Odds ratio is on the favorable side. So why not Oxaliplatin being used in these patients more than cisplatin.

Furthermore, In table 3, multivariate analysis depicts a marked decrease in OR of esophageal cancer as compared to table 2. This also needs explanation. 

Author Response

Thank you for your helpful comments. We had thoroughly reviewed the complete manuscript and carefully responded to all the comments. Our point-by-point responses are listed below.

Comments and Suggestions for Authors
This study address an important issue of chemotherapy induced hyponatremia. As acknowledged by the authors themselves, there are some limitations.

1. Additionally, in my opinion, the readership would be interested to know more about the ROR of cisplatin and Oxaliplatin in table 1. It is well known that either Cisplatin or Oxaliplatin can be used (with capecitabine) for esophageal cancer chemotherapy. And Oxaliplatin Odds ratio is on the favorable side. So why not Oxaliplatin being used in these patients more than cisplatin. 

Response:
Thank you for your valuable comments. As pointed out, oxaliplatin is a useful anticancer agent for esophageal cancer and is used worldwide. However, in Japan, the indication for oxaliplatin for esophageal cancer was added in 2019, and the number of cases used is small. Therefore, the analysis could not be performed due to the small number of reports in JADER, which was an adverse reaction report in Japan. Accordingly, we have now modified the discussion section as shown below:

Discussion (page 7, lines 223–227)
Also, oxaliplatin is a useful anticancer agent for esophageal cancer and is used worldwide [21]. However, in Japan, the indication for oxaliplatin for esophageal cancer was added in 2019, and the number of cases used is still small. Therefore, the analysis could not be performed due to the small number of reports in JADER, which was an adverse reaction report in Japan.

References
21.    Cunningham, D.; Starling, N.; Rao, S.; Iveson, T.; Nicolson, M.; Coxon, F.; Middleton, G.; Daniel, F.; Oates, J.; Norman, A.R. Upper Gas-trointestinal Clinical Studies Group of the National Cancer Research Institute of the United Kingdom. Capecitabine and ox-aliplatin for advanced esophagogastric cancer. N Engl J Med 2008, 358, 36–46. doi: 10.1056/NEJMoa073149.

2. Furthermore, in table 3, multivariate analysis depicts a marked decrease in OR of esophageal cancer as compared to table 2. This also needs explanation. 

Response:
Thank you for your helpful feedback. We have reviewed the suggestion you pointed out. Thirteen of the 17 patients with esophageal cancer were over 60 age, and 15 patients received cisplatin in table 3. Therefore, we considered that age (over 60) and cisplatin administration affected the results of multivariate analysis and that a marked decrease in OR of esophageal cancer was observed. Appropriately, we have included a sentence regarding this in the discussion section, as shown below:

Discussion (page 7, lines 241–246)
Further, in our study, table 2 presents a higher significant OR difference in esophageal cancer; however, multivariate analysis depicted a marked decrease in the OR of esophageal cancer. Thirteen of the 17 patients with esophageal cancer were over 60 age, and 15 patients received cisplatin in multivariate analysis. Therefore, age (over 60 age) and cisplatin administration might have contributed to the decreased OR of esophageal cancer in multivariate analysis.

Reviewer 2 Report

1. Line 49, 50: "The symptoms of hyponatremia are similar to those of chemotherapy side effects, such as fatigue, nausea, and somnolence; therefore, the symptoms of hyponatremia may be missed." Then how did the authors correlate or establish whether it was hyponatremia or side effects of anticancer drugs?  Was it reported within the database? 2. Please explain about hyponatremia in detail in the introduction part. 3. Line 36. How is hyponatremia related to seizure? Already we know an excessive discharge occurs when there is a case of seizure. 4. Please add a list of abbreviations within the manuscript. 5. Please Explain Figure 1 in details 6. The conclusion part should be more comprehensive.

Author Response

Thank you for your helpful comments. We had thoroughly reviewed the complete manuscript and carefully responded to all the comments. Our point-by-point responses are listed below.

Comments and Suggestions for Authors
1.     Line 49, 50: "The symptoms of hyponatremia are similar to those of chemotherapy side effects, such as fatigue, nausea, and somnolence; therefore, the symptoms of hyponatremia may be missed." Then how did the authors correlate or establish whether it was hyponatremia or side effects of anticancer drugs?  Was it reported within the database? 
Response:
Thank you for your helpful comments. As you have pointed out, the symptoms of hyponatremia are similar to other chemotherapy side effects. Therefore, we thought it was necessary to distinguish it from other side effects. However, cases in the JADER database were reported spontaneously, and it does not provide quantitative data on hyponatremia. Therefore, for this study, using Medical Dictionary ver. 24.1, we defined hyponatremia as a case of reported PT codes associated with hyponatremia (supplementary data), and we have added the following statement as a limitation of this study.
Discussion (page 8, lines 271–274)
Fourth, JADER does not provide the clinical information necessary to diagnose adverse events and underlying diseases, such as laboratory values. Therefore, in this study, we defined the diseases from the names of diseases reported spontaneously using the Dictionary of Medical Science ver. 24.1.

2. Please explain about hyponatremia in detail in the introduction part. 
Response:
Thank you for your insightful comments. As you have pointed out, we also agreed that there was a lack of explanation regarding hyponatremia. Thus, we have revised the introductory paragraph and included the appropriate references as follows:

Introduction (page 1, lines 31–54)
Hyponatremia is a critical electrolyte imbalance, which is associated with high mor-bidity and mortality rates [1]. However, hyponatremia may be asymptomatic if it is mild or slowly progressive. Worsening hyponatremia increases the risk of physical instability, falls, and inattention and results in central nervous system symptoms, such as disorienta-tion and seizures. These effects can reduce the patient’s quality of life and interfere with treatment, and older adults are more likely to be severely affected. Common causes of hy-ponatremia include medications and the syndrome of inappropriate antidiuretic hormone (ADH) secretion. Hyponatremia can be induced by several drugs, particularly some anti-cancer drugs [2]. The normal serum sodium concentration is generally 136–143 mEq/L, and a level <135 mEq/L is defined as hyponatremia. Hyponatremia occurs when there is an excess of total body water in relation to total sodium in the body. Because the state of extracellular fluid volume reflects the total sodium in the body, the state of extracellular fluid volume (decreased, normal, or increased) should be considered in hyponatremia. Hyponatremia caused by syndrome of inappropriate ADH or anticancer drugs is often associated with a normal extracellular fluid volume. Some previous studies have reported hyponatremia associated with the use of certain anticancer drugs, but the number of re-ported cases is limited since hyponatremia caused by anticancer drugs is rare [3–5]; therefore, knowledge regarding hyponatremia due to anticancer drugs is limited. The symptoms of hyponatremia are similar to those of chemotherapy side effects, such as fa-tigue, nausea, and somnolence; therefore, the symptoms of hyponatremia may be missed. It is thus important to anticipate the occurrence of hyponatremia in order to avoid the ad-verse effects and to manage it appropriately.

Hyponatremia is an important electrolyte imbalance associated with high morbidity and mortality [1]. Normal serum sodium concentrations range between 136 and 143 mEq/L, with hyponatremia defined as less than 135 mEq/L. As hyponatremia worsens, the risk of physical instability, falls, and inattention increases, including central nervous system symptoms, such as impaired consciousness and seizures caused by cerebral edema and increased intracranial pressure, may occur [2]. These effects reduce the patient's quality of life and interfere with treatment. In more severe cases, symptoms such as headache, lethargy, and jaundice may occur, eventually leading to seizures, coma, and death [3]. The incidence in hospitalized patients is reported to be 15–30% [4,5] and is known to be higher in the elderly with cirrhosis and heart failure [6,7].
Hyponatremia occurs when the total body water content is excessive in relation to the total sodium content in the body. Common causes of hyponatremia include medications and antidiuretic hormone (ADH) deficiency syndrome. Also, some drugs can induce hyponatremia, especially some anticancer agents [8]. Hyponatremia due to ADH incompatibility syndrome or anticancer agents is often associated with normal extracellular fluid volume. Accordingly, the status of extracellular fluid volume (decreased, normal, or increased) should be considered in hyponatremia because it reflects the total sodium in the body. Although hyponatremia associated with the use of certain anticancer agents has been reported in previous studies, re-reported cases are limited because hyponatremia due to anticancer agents is rare [9–11]; therefore, vast knowledge of hyponatremia due to anticancer agents is limited. Further, symptoms of hyponatremia are similar to those of chemotherapy side effects, such as fatigue, nausea, and somnolence, of which symptoms of hyponatremia may be missed. Therefore, it is important to anticipate the occurrence of hyponatremia and to treat it appropriately in order to avoid adverse effects.

References
1.    Spasovski, G.; Vanholder, R.; Allolio, B.; Annane. D.; Ball, S.; Bichet, D.; Decaux, G.; Fenske, W.; Hoorn, E.J.; Ichai, C.; Joannidis, M.; Soupart, A.; Zietse, R.; Haller, M.; van der Veer, S.; Van Biesen, W.; Nagler, E. Hyponatraemia Guideline Development Group. Clinical practice guideline on diagnosis and treatment of hyponatraemia. Eur J Endocrinol 2014, 170, G1-47. doi: 10.1530/EJE-13-1020.
2.    David, R.B.; Theodore, W. Post Clinical physiology of acid-base and electrolyte disorders. McGraw-Hill, 2001.
3.    Rondon-Berrios, H.; Agaba, E.I.; Tzamaloukas, A.H. Hyponatremia: Pathophysiology, classification, manifestations and management. Int Urol Nephrol 2014, 46, 2153–2165. doi: 10.1007/s11255-014-0839-2.
4.    Burst, V. Etiology and Epidemiology of Hyponatremia. Front Horm Res 2019, 52, 24–35. doi: 10.1159/000493234. 
5.    Davila, C.D.; Udelson, J.E. Hypervolemic hyponatremia in heart failure. Front Horm Res 2019, 52, 113–129. doi: 10.1159/000493242.
6.    Solà, E.; Ginès, P. Hypervolemic hyponatremia (liver). Front Horm Res 2019, 52, 104–112. doi: 10.1159/000493241.

3. Line 36. How is hyponatremia related to seizure? Already we know an excessive discharge occurs when there is a case of seizure. 
Response:
Thank you for your comments. Seizures are caused by cerebral edema and increased intracranial pressure. The difference in effective osmotic pressure between the brain and plasma causes brain cells to swell as water moves from outside to inside the cells. This occurs when hyponatremia progresses rapidly, and the brain has had too little time to adapt to the hypotonic environment. Accordingly, we have revised the statement and added appropriate references as follows.

Introduction (page 1, lines 33–36)
As hyponatremia worsens, the risk of physical instability, falls, and inattention increases, including central nervous system symptoms such as impaired consciousness and seizures, caused by cerebral edema and increased intracranial pressure, may occur [2].
References
2. Spasovski, G.; Vanholder, R.; Allolio, B.; Annane, D.; Ball, S.; Bichet, D.; Decaux, G.; Fenske, W.; Hoorn, E.J.; Ichai, C.; Joannidis, M.; Soupart, A.; Zietse, R.; Haller, M.; van der Veer, S.; Van Biesen, W.; Nagler, E. Hyponatraemia Guideline Development Group. Clinical practice guideline on diagnosis and treatment of hyponatraemia. Eur J Endocrinol 2014, 170,G1–47.

4. Please add a list of abbreviations within the manuscript.
Response: 
Thank you for your comments. We have added a list of abbreviations in the manuscript as follows:

Abbreviations (page 8, lines 300–303)
Abbreviations: JADER, Japanese Adverse Drug Event Report; ADH, antidiuretic hormone; AEs, adverse events; PMDA, Pharmaceuticals and Medical Devices Agency; ROR, reporting odds ratio; IQR, interquartile range; WSPs, Weibull shape parameters; CI, confidence interval; AVP, arginine vasopressin

5. Please Explain Figure 1 in details.
Response:
Thank you for your comments. As you have pointed out, we agreed with your suggestion that the explanation in Figure 1 was insufficient. Therefore, we revised the figure and figure legends as follows:

Figure legends (page 6, lines 198–202)
Figure 1. The number of hyponatremia cases associated with cisplatin by onset time in the Japanese Adverse Drug Event Report (JADER) database.

Figure 1. The number of hyponatremia cases associated with cisplatin by onset time in the Japanese Adverse Drug Event Report (JADER) database.
It showed the time-to-onset profiles of hyponatremia following cisplatin administration. The onset of hyponatremia was analyzed using the Weibull distribution, wherein the scale parameter α of the Weibull distribution determines the scale of the distribution function. The WSP β of the Weibull distribution was employed to show the hazard level over time without a reference population. 

7.    The conclusion part should be more comprehensive.
Response:
Thank you for your comments, and we have enriched the conclusion part, as you suggested, which is shown below:

Conclusions (page 8, lines 277–286)

This study suggested that hyponatremia may more likely occur in patients with lung cancer and those who were administered cisplatin. Based on this information, patient information should be carefully considered before administration, monitoring during treatment, and appropriate measures should be applied for patients requiring early sodium supplementation.

 ↓

This study suggested that the anticancer drugs (cisplatin, etoposide, and pemetrexed) and cancer types (esophageal, lung, and renal cancers) had hyponatremia signals, and hyponatremia may more likely occur in patients with lung cancer and those who were administered cisplatin. In addition, hyponatremia, which is associated with cisplatin treatment, was detected in women and patients aged >60 years. Furthermore, the onset of hyponatremia, which is associated with cisplatin treatment, developed relatively early (within 10 days). Based on this information, patient information should be carefully considered before administration, monitoring during treatment, and appropriate measures should be applied for patients requiring early sodium supplementation.
